# Second-Hand Tobacco Smoke Exposure and Smoke-Free Environments in Ethiopia: A Scoping Review and Narrative Synthesis

**DOI:** 10.3390/ijerph19148404

**Published:** 2022-07-09

**Authors:** Terefe Gelibo Argefa, Selam Abraham Kassa, Noreen Dadirai Mdege

**Affiliations:** 1ICAP in Ethiopia, Mailman School of Public Health, Columbia University, Addis Ababa P.O. Box 5566, Ethiopia; mamater.1986@gmail.com; 2Development Gateway: an IREX Venture, 1100 13th St NW, Suite 800, Washington, DC 20005, USA; skassa@developmentgateway.org; 3Department of Health Sciences, University of York, York YO10 5DD, UK; 4Centre for Research in Health and Development, Amos Drive, York YO42 2BS, UK

**Keywords:** smoking, tobacco smoke exposure, second-hand smoke, smoke-free environments

## Abstract

Ethiopia passed a law prohibiting tobacco smoking in all public places in 2019. We conducted a scoping review to identify gaps in the existing literature on second-hand smoke (SHS) exposure and smoke-free environments in Ethiopia that need to be prioritised for future research to support policy and practice. We conducted systematic searches in January 2022 in the following databases: Medline, EMBASE, and PsycInfo. Two reviewers independently screened the identified study reports for eligibility and extracted data from the eligible studies. The extracted data was descriptively analysed, and research recommendations were drawn. A stakeholder consultation workshop was held to identify research topics on SHS exposure and smoke-free environments in Ethiopia that they perceived to be priorities for primary research. Of the 388 research reports identified, only nine were included in the scoping review. The topics explored includes prevalence of SHS exposure (six studies); knowledge on SHS exposure (three studies); compliance to smoke-free environments legislation (two studies); and exposure to anti-smoking messages (one study). The stakeholders prioritised further research addressing compliance monitoring and enforcement of the smoke free laws in Ethiopia. There is a need for studies that test new methods for compliance monitoring and enforcement, evaluate strategies to increase knowledge on the harms of SHS exposure and the smoke-free legislation, and evaluate the current smoke-free legislation in Ethiopia.

## 1. Introduction

Overall, 3.7% of adults in Ethiopia (6.2% of males, and 1.2% of females) smoke tobacco [1]. Exposure to second-hand tobacco smoke (SHS) results in 1.2 million deaths worldwide each year [2], with 47% and 28% of these deaths occurring in women and children, respectively [3]. Most of the deaths associated with SHS exposure are in low-income and middle-income countries (LMICs) [3]. Among adults in Ethiopia, 29.3% (6.5 million) of those who worked indoors reported that they were exposed to SHS in their workplace, and 12.6% (8.4 million) were exposed at home in 2016 [4]. In the same year, 60.4% of people visiting bars and nightclubs reported being exposed to SHS in these settings; and this was 31.1% in restaurants, 19.7% in government buildings, 11.4% in public transportation and 7.0% in health-care facilities [4]. SHS exposure among children is also high. In the last Global Youth Tobacco Survey (GYTS) among students aged between 13 and 15 years in Addis Ababa, about 29% lived in homes where others smoked in their presence; and over 40% were exposed to tobacco smoke in public places [5]. In a recent study among 1673 adolescents aged 13–19 years in Hawasa and Jimma town, 17% of non-smokers were exposed to SHS in their homes, and 61% were exposed in public places [6].

Ethiopia’s Tobacco Control Strategic Plan 2010–2012 Ethiopian Calendar (E.C). (2017/18–2019/20) recognised SHS exposure as a public health threat, and made a commitment to enact and enforce smoke-free public places and environments [7]. In 2019, Ethiopia joined a group of 69 countries worldwide that provide protection from SHS exposure at best-practice level, with all public places completely smoke-free [8]. The country passed a law prohibiting smoking in all public places, i.e., any part of all indoor workplaces, all indoor public places, all means of public transport, and all common areas within condominium housings (Proclamation No.1112/2019) [9]. In addition, smoking in any outdoor part of healthcare facilities, government institutions, facilities including schools intended mainly for those under the age of 21, higher education institutions, youth centres, and amusement parks is prohibited. The law also bans smoking or tobacco use in any indoor and outdoor space within 10 metres of any doorway, operable window, or air-intake mechanism of any public place or workplace [9]. Enforcement of the smoke-free regulations has been scaled-up through awareness raising and compliance inspections across the country: more than 16,000 inspections of a variety of public places across the country have been reported since 2019 [8].

Reducing SHS remains a top priority for the Ethiopian government [10]. Tobacco control stakeholders in Ethiopia, including government ministries, civil society organisations and academic institutions agree that there are some glaring gaps in evidence on the topic to inform policy and practice [10]. However, there has not been an attempt to bring together the currently existing evidence on SHS exposure and smoke-free environments in Ethiopia in order to identify specific research questions that need to be prioritised. We therefore conducted a scoping review and narrative synthesis aimed at examining the extent, range, and nature of research activity; summarising the findings from the studies identified; identifying gaps in the existing literature on SHS exposure and smoke-free environments in Ethiopia; and identifying research topics on SHS exposure and smoke-free environments in Ethiopia that need to be prioritised for primary research intended to support policy and practice.

## 2. Methods

### 2.1. Study Design

We utilised a well-established scoping review methodology comprising identifying relevant studies, selecting studies to include in the review, charting the data, collating, and summarising and reporting the results [11]. We also conducted a consultation exercise where we invited tobacco control stakeholders in Ethiopia who were not part of the review team to provide their insights and validate findings from the scoping review.

### 2.2. Study Identification

We conducted systematic searches in the following electronic databases using a comprehensive search strategy that was developed iteratively through a literature review and reviewing the results of initial scoping searches: Medline (1946 to 18 January 2022), EMBASE (1974 to 18 January 2021) and PsycInfo (1806 to January Week 2 2022). Our search strategy combined SHS exposure/smoke-free environment terms with Ethiopia (Appendix A). The following keywords were used for the searches: second-hand smoke, passive smoke, environmental exposure, environmental tobacco smoke, cigarette smoke, smoke-free, and Ethiopia. In addition to searching electronic databases, we also looked through the reference lists of included studies and consulted stakeholders and researchers working on the topic in Ethiopia to identify any study reports that we might have missed including grey literature. The stakeholders consulted included representatives from the Ministry of Health and other governmental and non-governmental institutions working on this topic in order to identify any relevant government reports or those from other stakeholders. No date, study design, publication type or language limits were imposed on the searches.

We followed the Cochrane and Centre for Reviews and Dissemination guidelines in designing searches [12,13]. The search strategies were reviewed by two reviewers who were not involved in building the search using the Peer Review of Electronic Search Strategies checklist [14]. The searches were reported as per the Preferred Reporting Items for Systematic reviews and Meta-Analyses (PRISMA) literature search extension [15].

### 2.3. Study Selection

We used the Rayyan software (Rayyan—Intelligent Systematic Review https://www.rayyan.ai/ (accessed on 5 January 2022)) to manage and screen the research reports retrieved from the searches. Each study was independently screened for eligibility by two reviewers using a standardised, study selection form based on the prespecified study eligibility criteria. The form was piloted on five studies before use. The screening was done in two stages: screening of titles and abstracts of all retrieved records after removing duplicates; and then screening of the full texts of those studies identified in the first stage as potentially eligible. Any disagreements regarding study inclusion were resolved through discussion between the two reviewers.

Studies were eligible if they were carried out in Ethiopia anytime up to 18 January 2022, and addressed any aspect of SHS exposure, i.e., exposure to “the smoke emitted from the burning end of a cigarette or from other tobacco products usually in combination with the smoke exhaled by the smoker” [16], or smoke-free environments, i.e., environments where the air is 100% free from tobacco smoke (including, but is not limited to, air in which tobacco smoke cannot be seen, smelled, sensed or measured) [16] in Ethiopia. Studies that were not eligible if they were only available as an abstract.

### 2.4. Data Charting

A data charting form developed in Microsoft Excel (Office 2019) specifically for the review and based on the data domains and variables listed in Table 1 was used to chart the data from the included studies. The data extraction tables were accompanied by instructions and decision rules for coding data in order to increase consistency, reduce bias from subjective judgement and improve the validity and reliability of the process [13,17]. One researcher read each included study in full, putting any data that is relevant to the review under the appropriate data domains. For studies that were reported in more than one publication, we charted the data from each publication separately and then combined the data. A second reviewer checked the charted data for accuracy and quality. We also extracted main study results as reported in each study. Disagreements between reviewers were resolved through discussion and consensus.

### 2.5. Data Collating, Summarising and Reporting

We conducted content analysis to identify and categorise various data items, and produced tables, graphs and narrative summaries that mapped the extent, range, and nature of research activity according to the domains and variables described in Table 1. We used these summaries to identify the key gaps in the existing literature and make recommendations for future research. We also narratively synthesised study findings according to their study focus areas. The reporting of the scoping review followed the PRISMA extension for scoping reviews [18].

### 2.6. Stakeholder Consultations

We held an online, one and a half hour, stakeholder consultation workshop to share and validate our findings. Specifically, we utilised this stage to identify, from the perspective of the stakeholders, after they had been presented with the scoping review results, what the priority research topics on SHS exposure and smoke-free environments in Ethiopia are. First, we presented the objectives, methods and results of the scoping review, and a preliminary list of recommendations that we had drawn from this literature (15 min). The stakeholders were given a chance to ask questions and seek clarification on any aspect of the presentation; and to discuss the recommendations amongst themselves, adding and removing recommendations from the preliminary list as they saw fit (30 min). We did not prescribe specific criteria for the discussions, but requested them to include the following as part of their considerations: the importance of the topic for policy making in Ethiopia; alignment with the government priorities; and the feasibility of conducting research on the topic, for example, in light of the current political climate and other resource constraints. After finalising the research recommendations list, the stakeholders were given one to two minutes to independently vote for their first choice in real time through a poll created on Zoom. The invited stakeholder organisations included academic and research institutions, civil society organisations, the World Health Organisation country office, the Ethiopia Food and Drug Administration, and relevant departments in the Ministry of Health. The list is provided in Appendix A.

## 3. Results

### 3.1. Search Results

The database searches yielded 384 records and 366 remained after duplicates were removed. Of the 366 articles, the five studies that were eligible for full-text review were all found to be eligible for inclusion. Three additional eligible studies were identified from the reference list searches, and one more study was identified through stakeholder consultations. Therefore, a total of nine studies were included in the scoping review (Figure 1) [6,19,20,21,22,23,24,25,26].

### 3.2. General Characteristics of Studies

All of the included studies were cross-sectional surveys (Table 2) [6,19,20,21,22,23,24,25,26]. Four studies were conducted in Southern Ethiopia [6,22,23,24], whilst there were three studies in Central Ethiopia [19,26], and one each for East [25], West [21] and South West Ethiopia [6]. One study used data from a nationwide survey covering all 11 administrative districts [20]. Of the nine studies, there were two studies each for Addis Ababa [19,26], Alata Wondo [23,24] and Hawasa [6,22], and one each for Jimma [6], Nekemte [21] and Kersa towns [25]. Four of the studies were household level surveys [20,23,24,25], whilst two were conducted in schools [6,21], and one was at a university [22]. The remaining study covered a number of settings: food and beverage providing facilities; health care facilities; schools; youth centres; and government offices [19].

Sample sizes ranged from 221 to 10,150. All of the included studies were published between 2013 and 2022. Eight of the nine studies used data that was collected sometime between 2010 and 2018 [6,20,21,22,23,24,25,26], which is before Ethiopia’s current federal tobacco control legislation (Proclamation 1112/2019) [9], and the 2021 Tobacco Control Directive No. 771/2021 [27]. The remaining study collected data in 2021 in Addis Ababa, but this was before launching an initiative to make the city smoke-free [19].

### 3.3. Study Focus

Overall, six of the nine studies reported prevalence of SHS exposure in public places or at home [6,21,22,23,24,25] one explored exposure to anti-tobacco messages [20]; three reported on knowledge on SHS exposure [20,24,25]; and two reported on compliance with smoke-free environments laws/regulations [19,26].

### 3.4. Study Population Characteristics

Two of the studies were conducted among school adolescents aged between 13 and 19 years (Table 3) [6,21]. The rest were conducted in adults: two among those ≥ 15 years of age [6,21]; two among those from 18 to 55 years of age [23,24]; one among university students aged from 20 to 24 years [22]; and one among hospital staff [26]. The study among hospital staff also conducted observations of 15 buildings located in the hospital premises of five hospitals that had started the implementation of smoke-free legislation [26]. Two of the adult sample studies had all female samples [23,24]. One study collected data from food and beverage providing premises, health service premises, schools, youth centres and government offices through observation [19].

### 3.5. Summary of Study Findings

The study results are summarised below by study focus areas.

#### 3.5.1. Prevalence of SHS Exposure

Exposure among young people: The prevalence of self-reported SHS exposure in public places among adolescents attending school was at least 60% [6,21]. Self-reported exposure at home was also high, ranging from 17% to 38% of participants [6,21]. In the study among university students, 69% of participants reported exposure within university campuses, with 17% reporting daily exposure [22]. Exposure was in almost all areas within the campuses, including in the university compound, dormitories, toilets, lounges, cafes and class rooms [22]. Petersen et al. found that in 5.1% of households young children were frequently exposed to SHS in the home [23].

In the study by Ababulgu et al., the following significantly increased the odds (three times) of being exposed to SHS at home: being female when compared to being male (Odds Ratio (OR) 3.46; 95% CI: 2.62–4.57), having parents (OR 3.34; 95% CI: 2.37–5.03) or closest friends (OR 3.61; 95% CI: 2.41–5.41) who smoke compared to those who do not [6]. The following significantly increased the odds of SHS exposure in public places: having a peer who smokes (OR 3.76; 95% CI: 2.49–5.65) compared to those who did not, and not being informed on the danger of health effects of tobacco smoke (OR 5.32; 95% CI 4.13–6.81) [6].

Exposure among adults: In two studies, 15% [23] and 40% [25] of households reported that smoking was allowed in the home. In one of these studies, an additional 12% of households allowed smoking in the home with exceptions, and 17% did not have rules on second hand smoke [25]. In the study by Petersen et al., 14.4% of participants reported that smoking took place daily in their homes [23,24], whilst this was 33% in the study by Reda et al. [25]. Petersen et al. reported that being an urban residents increased the likelihood to report that smoking was allowed in the home, or that SHS occurred daily in their household [23]. In addition, having a house member who is a current tobacco user, recent exposure to point-of-sale advertising, and living in a household where smoking is allowed in the home increased the likelihood of reporting daily occurrence of SHS in the home [23].

#### 3.5.2. Knowledge on SHS Exposure

Reda et al. [25] and Petersen et al. [24] both reported high proportions of respondents who thought/believed that SHS exposure was harmful (73% and 98%, respectively). Nevertheless, knowledge on specific health conditions was high among participants in the study by Reda et al. than Petersen et al.: 82% versus 8% for heart disease, 90% versus 20% for respiratory problems; and 84% versus 5% for lung cancer, respectively. These two studies were conducted in different towns, and one was 75% male [25], while the other was 100% female [24].

In the study by Bekalu et al. [20], although the risk perception was high overall, the proportion reporting that SHS exposure causes serious illness were higher in those who were more educated, wealthier, and those who were urban residents.

#### 3.5.3. Compliance with Smoke-Free Environments Laws/Regulations

In the study by Tadesse and Zawdie, only 2.8% of the inspected places in the five hospitals had anti-smoking signs, and this was more likely to be at entrances than other places [26]. Cigarette butts were observed in all hospitals and these were more likely to be found at the entrances (24%) and toilets (15%) than other places [26]. Actual smoking on hospital premises was observed in three of the five hospitals. Overall, 42% of hospital staff were aware of the hospital smoke-free legislation, whilst 34% were not aware and 24% were not sure that there was a smoke-free legislation that prohibited smoking within hospital premises. A small proportion of hospital staff (10%) reported not being compliant with smoke free legislation—the levels of knowledge about smoke-free legislation was poor among 69% of these staff, and 83% had unfavourable attitudes towards the legislation. In contrast, 42% of compliant hospital staff had poor knowledge of the legislation, and 47% had unfavourable attitudes towards it. Additionally, 28% of the hospital staff indicated that they had been enforcing the smoke-free legislation [26].

In the report by the Ethiopia Food and Drug Authority of their compliance monitoring exercise conducted in 2021 [19], overall compliance for the Bole and Arada sub-cities was low: 33% (4/12) for youth centres, 31% (8/26) for government offices, 54% (27/50) for health facilities, 45% (13/30) for educational facilities and 25% (26/103) for food and beverage venues [23]. Compliance was much higher in Arada when compared to Bole for most settings: 46% versus 16% for government offices, 67% versus 42% for health facilities and 68% versus 20% for educational facilities. Compliance was similar in food and beverage venues, 23% for Arada and 26% for Bole sub-city; and higher for Bole for youth centres (40% versus 0%). A number of challenges with compliance were reported including fear of conflict with clients, lack of client awareness of the legislation and cigarette sales within 100 m for the establishments. In addition, for food and beverage venues, there was a fear of losing customers, with foreign customers being treated differently (i.e., being allowed to smoke) to local ones. For schools, there were reports of people smoking along the school fence and also the presence of shisha houses in surrounding areas causing problems for compliance. Designated smoking areas were present in a few of the food and beverage venues and youth centres [19].

#### 3.5.4. Exposure to Anti-Smoking Messages

In the study by Bekalu and colleagues, the association between exposure to anti-smoking messages and risk perception of SHS exposure was not significant (OR 1.26; 95% CI: 0.64–2.46) [20].

### 3.6. Stakeholder Consultation Results

After analysing and synthesising the results, two reviewers developed a preliminary list of research recommendations to be used as a starting point for the discussions with the stakeholders (Table 4). The stakeholder consultation workshop was attended by 11 participants representing nine stakeholder organisations: two represented the Ministry of Health, one represented the Ethiopia Food and Drug Administration, three represented academic institutions (one each for the Addis Ababa University, Department of preventive medicine; Hawassa University, College of Medicine and Health Sciences; and Jimma University, College of Health Sciences & Medicine) and five participants were from CSOs (two for Mathewos Wondu Ethiopia Cancer Society, and one each for the Health Development and Anti Malaria Association, MeQuamia Community Development Organization, and Campaign for Tobacco Free Kids).

The stakeholders added ‘*Reducing levels of SHS exposure within households and communities*’ to the list of research recommendations. Eight out of the eleven stakeholders cast their votes for their preferred research priority in the poll. The recommendation with the majority of votes was compliance monitoring and enforcement, specifically evaluating the feasibility, acceptability, effectiveness, cost-effectiveness of alternative methods of compliance monitoring and enforcement (Figure 2). There were no votes for compliance challenges and developing and testing the effectiveness and cost-effectiveness of different strategies for awareness among those who should comply as well as the general population. These topics are therefore not represented in Figure 2.

## 4. Discussion

This scoping review highlights the scarcity of research on SHS exposure and smoke-free environments in Ethiopia. Of the nine identified studies, most (six) report on exposure to SHS either in the home or at academic institutions (i.e., schools and universities), or knowledge of harms due to SHS (three). Exposure in the other indoor public places, indoor workplaces and outdoor public places that are covered by the Ethiopian legislation have not been investigated in detail, except through the Global Adult Tobacco Survey which was last conducted in 2016 [4] and the GYTS conducted in 2003 [5]. Reports on the levels of compliance to legislation are scarce: only two were identified. One of these studies covered a wide range of settings, i.e., youth centres, government offices, health facilities, educational institutions and food and beverage venues, albeit only in two sub-cities in Addis Ababa [19]. All of the identified studies used data that was collected either before Ethiopia’s current federal tobacco control legislation (Proclamation 1112/2019) [9], and the 2021 Tobacco Control Directive No. 771/2021 [27], or before local implementation of smoke-free initiatives.

Studies included in this review reported high levels of SHS exposure in the home among adolescents and adults, and at universities among young people. Being female, having members of households who smoke, living in urban areas and poor knowledge of the dangers of SHS exposure significantly increased the likelihood of SHS exposure in these settings. In addition, living in a household where smoking inside the home was permitted increased the likelihood of SHS exposure within the home. These findings are in line with prior research that has shown that women are worst affected by the SHS exposure disease burden, with 47% of deaths from SHS exposure occurring in female adults [3]. A recent analysis of SHS exposure among adolescents aged 12–15 years across 68 LMICs also reported a positive association between SHS exposure and parental smoking, with the association being larger for maternal smoking than paternal smoking [28]. Another study in The Gambia also found that lower education attainment, lower household income, urban residence, and not being aware of smoke-free regulations increased the risk of SHS exposure [29].

The studies included in the review found low levels of compliance with smoke-free laws in different establishments. In all of the studies, lack of knowledge of the legislation among staff working in those establishments as well as their clients seemed to be one of the causes of non-compliance and negative attitudes towards the legislation. The presence of cigarette sales within the vicinity, fear of conflict with clients, and fear of losing clients and revenue (in hospitality venues) also seem to play a part in non-compliance. Sub-optimal levels of compliance with smoke-free policies within hospitality venues have also been observed in other African countries [30,31,32]. In Ghana and Uganda, poor knowledge of the smoke-free legislation among hospitality venue staff was also reported [31,33]. Nevertheless, in a number of countries, including in Nigeria and Ghana, the attitude towards smoke-free laws among hospitality staff has been found to be generally positive [31,34,35,36], although the fear of revenue reductions seems to be a consistent finding within these venues [35,36]. 

The studies suggest that there could be demographic as well as geographical differences in both knowledge of the harms of SHS exposure and compliance to legislation. The study by Bekalu et al. observed a social gradient on knowledge, with the proportion reporting that SHS exposure caused serious illness being higher in those who were more educated and wealthier [20]. This proportion was also higher in urban than rural residents [20]. Furthermore, the proportion of participants with good knowledge of specific conditions that are attributable to SHS exposure was much higher in the study by Reda et al., which was conducted in Kersa town in a sample with 75% men [25], when compare to those reported by Petersen et al. in a study conducted in Alato town in an all-female sample [23].

In addition to the identification of research priorities, the findings from our stakeholder consultation meetings also highlighted a spread on opinions on what the research priorities should be. This could be as a result of the different interests and focus areas among the stakeholders. We incorporated stakeholder discussion on research priorities in order to facilitate understanding of each other’s perspectives, the consideration of those different perspectives, and broad acceptance of the process and outcomes [37].

Reducing SHS exposure and increasing compliance to smoke-free laws remain a priority for Ethiopia. Increasing the knowledge and awareness of the harms of SHS exposure and the smoke-free laws among the public and staff working at different venues that need to be smoke-free seems to be important for reducing SHS exposure as well as increasing compliance. It could potentially reduce conflicts between staff within establishments and their clients and improve attitudes towards smoke-free legislation. Bekalu et al. suggests that mass media communication as a potential strategy for increasing knowledge and awareness of these issues [20]. Nevertheless, such a strategy would need to account for patterns of mass media use in Ethiopia, for example the types of mass media platforms commonly used and any sociodemographic (e.g., gender and age) or geographic (e.g., urban versus rural) differences that might exist in the access and use of those platforms [38,39,40].

The impact of the current smoke-free legislation on SHS exposure and compliance needs to be evaluated, for example using before and after comparisons where data allows. The current phased implementation of the legislation also offers an opportunity for natural experiments on the impact of the legislation on both exposure and compliance. Continuous evaluation of tobacco control policies could identify the gaps and promote the development of new strategies for improving compliance with the laws [41]. There is also a need to develop and test the effectiveness and cost-effectiveness of different strategies of increasing knowledge of the harms of SHS exposure and on the smoke-free laws. In addition to the use of mass media, other strategies could include workplace-, educational facility-, and community-based interventions. For example, educational facility-based interventions targeting children and young people, including teaching them to negotiate for smoke-free environments, or community-based interventions for example via faith-based institutions hold promise, although most of studies exploring them have mainly been in the context of exposure within the home [42,43,44]. The concerns around loss of revenue in hospitality settings as a result of implementing smoke-free policies, need to be explored further. In addition, research should develop and test strategies to overcome the other challenges raised by staff members at different establishments such as cigarette sales within 100 m of the establishments and smoking within the vicinity of educational establishments. Another potential area of research is the evaluation of the feasibility, acceptability, effectiveness, cost-effectiveness of alternative methods of compliance monitoring and enforcement.

Compliance monitoring in Ethiopia and other countries worldwide is currently mainly through visual inspection of a venue using standard compliance indicators such as observed smoking; presence of designated smoking areas, ashtrays, no smoking signs, cigarette butts or the smell of tobacco smoke at the venues [45]. This method is attractive because it is relatively simple and inexpensive [45]. However, some studies have highlighted the potential usefulness of low-cost air quality monitors, such as those that measure the concentration of airborne particulate matter less than 2.5 microns (PM_2.5_) in diameter. For example, in a study in Uganda, although the occurrence of at least one observational indicators of smoking within the 222 hospitality venues was in less than 50% of the venues, the average PM_2.5_ for a subsample of 108 study sites was 171 µg/m^3^, representing an unhealthy air quality rating [30]. The average rating in indoor venues indicated hazardous levels (PM_2.5_ concentration of 267.64 µg/m^3^), whilst outdoor venues had an average concentration of 85.60 µg/m3 indicating unhealthy air quality [30]. The results from these monitors can also be translated into visual aids that can be used to provide real-time feedback on air quality in a venue, including the implications of the recorded levels to health: this has been shown to be an effective educational and behaviour change motivational tool for reducing SHS exposure [46]. The majority of tobacco control stakeholders in Ethiopia who attended our workshop in fact voted this topic as their top priority.

### Limitations of the Study

We used a comprehensive list of search terms. The searches were conducted in three of the main bibliographic databases for health-related research. Although attempts were made to identify grey literature through contacting key stakeholders, it is still possible that some of this literature was missed. Almost all the included studies were conducted before the implementation of Tobacco Control Directive; therefore, the findings may not be indicative of the implementation of the provisions of the current smoke free laws in Ethiopia. We gave stakeholders up to two minutes to independently vote for their top research priorities in real time during the meeting. This might have limited their ability to reflect on the discussions and gather more information to enable them to fully consider their choices. This approach however enabled us to maximise the number of responses obtained.

## 5. Conclusions

Our scoping review highlighted the scarcity of research on SHS exposure and smoke-free environments in Ethiopia. As prioritised by the stakeholders, further research is needed to address compliance monitoring and enforcement, particularly evaluating the feasibility, acceptability, effectiveness, cost-effectiveness of alternative methods of compliance monitoring and enforcement (e.g., objective measures with visual aids that can be used to provide real-time feedback on air quality in a venue), including the implications of the recorded levels to health.

## Figures and Tables

**Figure 1 ijerph-19-08404-f001:**
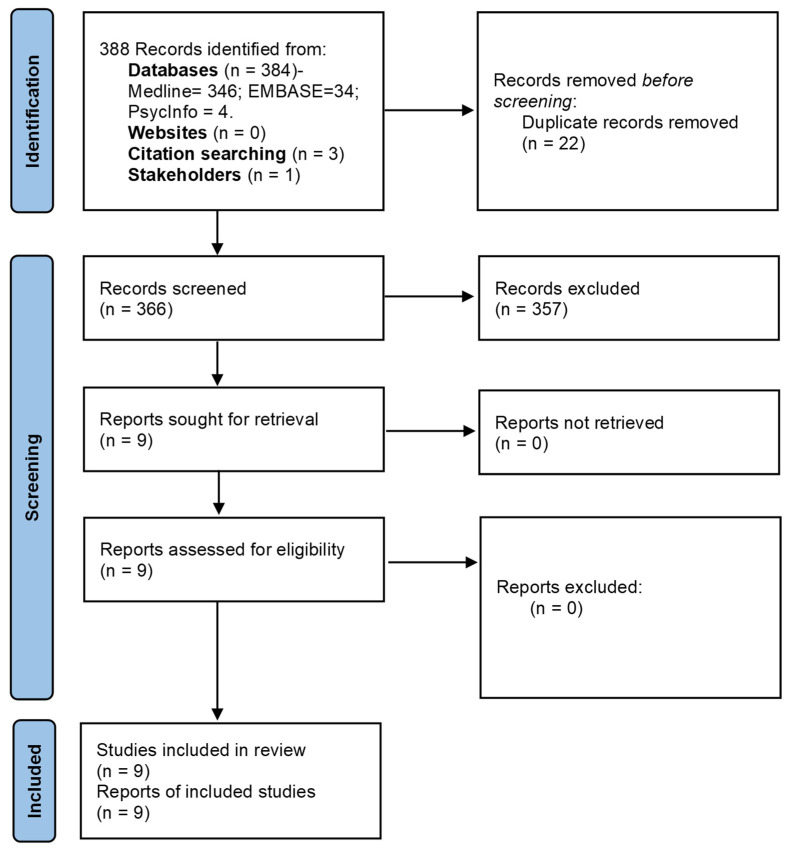
Flow of studies in the review process.

**Figure 2 ijerph-19-08404-f002:**
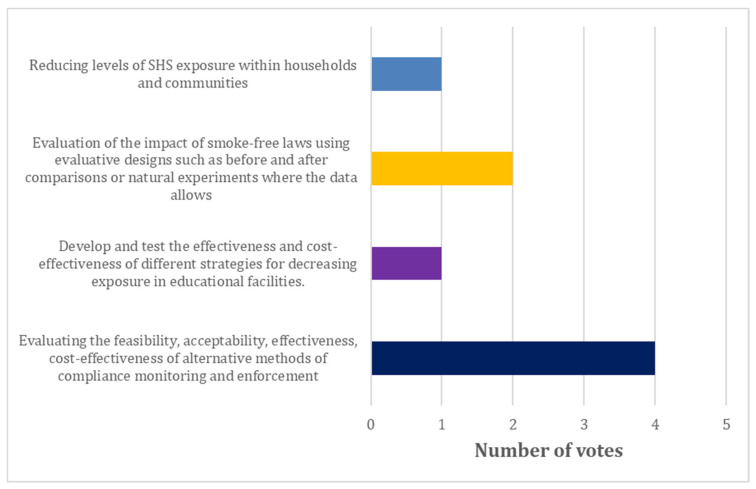
Stakeholder poll results.

**Table 1 ijerph-19-08404-t001:** Data extraction domains and variables.

Domain	Variable
General study characteristics	Author, year of publication, geographic location (i.e., administrative district), study setting (e.g., school, health care facility, bar, restaurant, hotel, etc.), study design, sample size
Principal focus	Topic (e.g., exposure, enforcement, compliance, awareness, etc.), research question(s), type(s) of intervention(s) if applicable
Study population characteristics	Age, gender, ethnicity, socio-economic status

**Table 2 ijerph-19-08404-t002:** General study characteristics.

Author, Year	Geographical Region	City/Town	Study Setting	Sample Size
Ababulgu et al., 2016 [6]	Southern and South West	Hawassa and Jimma towns	Schools	1704
Bekalu et al., 2022 [20]	Nation wide	All the 11 administrative regions	Households	10,150
Bobo et al., 2018 [21]	Western	Nekemte town	Schools	1088
Ethiopia Food and Drug Authority, 2021 [19]	Central	Addis Ababa (Bole and Arada sub-cities)	Food and beverage providing facilities; health care facilities; schools; youth centres; and government offices	221 (103 Food and Beverage Service Providers, 50 health service providers, 30 schools, 12 Youth Centres and 26 government offices included)
Kassa and Deyno, 2014 [22]	Southern	Hawassa town	University	590
Petersen et al., 2016 [23]	Southern	Alata Wondo town	Households	372
Petersen et al., 2018 [24]	Southern	Alata Wondo town	Households	372
Reda et al., 2013 [25]	Eastern	Kersa town	Households	600
Tadesse and Zawdie, 2019 [26]	Central Area	Addis Ababa	Hospitals	354

**Table 3 ijerph-19-08404-t003:** Study population characteristics.

Author, Year	Population Group	Age Range Mean Age (Standard Deviation)	Gender
Ababulgu et al., 2016 [6]	Adolescents enrolled in grade 9 to 12	13–19 years Not reported	47.7% male; 52.3% female
Bekalu et al., 2022 [20]	Adults	≥15 years 31.2 (14.8) years	49.9% male; 50.1% female
Bobo et al., 2018 [21]	Adolescents attending high school	14–19 years Not reported	57.9% male; 42.1% female
Ethiopia Food and Drug Authority, 2021 [19]	Food, beverage and health service providers, schools, youth centres and government offices	-	-
Kassa and Deyno, 2014 [22]	Adults (University students)	20–24 years 20.7 (1.49) years	81.7% male; 18.3% female
Petersen et al., 2016 [23]	Adults	18–55 years 29.4 (6.9) years	100% female
Petersen et al., 2018 [24]	Adults	18–55 years 29.4 (6.9) years	100% female
Reda et al., 2013 [25]	Adults	≥15 years 35 (15.0) years	75.1% male; 24.9% female
Tadesse and Zawdie, 2019 [26]	Adults (hospital staff)	29 years * (Interquartile range 26 to 35 years)	54.3% male; 45.7% female

* Median.

**Table 4 ijerph-19-08404-t004:** Research recommendations drawn from the literature.

Topic	Potential Research Objectives/Questions
Compliance monitoring and enforcement	Evaluation of the feasibility, acceptability, effectiveness, cost-effectiveness of alternative methods of compliance monitoring and enforcement, e.g., objective measures with visual aids that can be used to provide real-time feedback on air quality in a venue, including the implications of the recorded levels to health.
Compliance challenges	Investigate the impact of implementing smoke-free policies in hospitality venues on revenue. Develop and test strategies to overcome the other challenges such as cigarette sales within 100 m for the establishments and smoking within the vicinity of educational establishments.
Knowledge of the harms of SHS exposure	Develop and test the effectiveness and cost-effectiveness of different strategies for increasing knowledge of the harms of SHS exposure.
Awareness of the smoke free law	Develop and test the effectiveness and cost-effectiveness of different strategies for increasing Awareness of the smoke free law.
Exposure in educational facilities	Develop and test the effectiveness and cost-effectiveness of different strategies for decreasing exposure in educational facilities.
Impact of the smoke-free law	Evaluation of the impact of smoke-free laws using evaluative designs such as before and after comparisons or natural experiments where the data allows.

## Data Availability

The data supporting the findings of this review are available within the paper and its Appendix A.

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
