# Peer review of "Second-Hand Tobacco Smoke Exposure and Smoke-Free Environments in Ethiopia: A Scoping Review and Narrative Synthesis"

_ijerph, 2022, doi:10.3390/ijerph19148404_

Round 1
Reviewer 1 Report
While there is a large volume of research on tobacco smoke in different parts of the world, this review study does add new insights to the current knowledge on the diverse situation of tobacco smoke in a specific country.
The paper is generally well written. However, there are improvements the authors can make to enhance the quality of the paper.
Please provide the full name of SHS when the term first appears. Line 11
Please provide the overall smoking prevalence in Ethiopia
“~29% lived in homes where others smoke” line 38. Change “ ~ “into “about.”
“Ethiopia’s Tobacco Control Strategic Plan 2010-2012 E.C.” Line 42. What does E.C mean?
“We contacted authors of the studies by email, where data was not available or was unclear”. Line 117. How many studies’ authors were contacted?
“There were two studies each in Addis Ababa [18, 26], Alata Wondo [22, 23] and Hawasa [5, 21]” Line 163. How many studies, according to the references?
“All of the included studies were published 169 between 2013 and 2022 inclusive.” Line 170. Please delete“inclusive”
“Overall, six of the nine studies reported prevalence of SHS exposure [5, 20, 21, 23, 24, 177 27], one explored exposure to anti-tobacco messages [19], three reported on knowledge 178 on SHS exposure [19, 23, 24], and two reported on compliance with smoke-free environments laws/ regulations (Figure 2) [18, 26].” Lines 177-180. Change some commas into semicolons.
Delete Figure 2 because the data illustrated in the figure were already described in the text.
“One study was an observational study in food and beverage providing premises…”Line 190. At the same time, the author claimed that“All of the included studies were cross-sectional surveys”. Please make clear the design type of the included study.
In the discussion section, please explain the findings from the stakeholders’ opinions.
The authors had better re-structure paragraphs in the discussion section. Please move the paragraph beginning with“We used a comprehensive list of search term…”to the place above the Conclusions section. Please add the subheading “ limitations of the study” to this paragraph.
Please add a subheading“ Practical implications”to the section from Lines 356-403.
Reviewer 2 Report
The authors state that “Our review highlights an urgent need for evidence to inform policy and practice on SHS exposure and smoke-free environments in Ethiopia.”
They do not say what kind of studies should be carried out to assess compliance with the law.
The team of Esteve Fernandez and Maria José López, from Barcelona, has carried out many studies in this area that can serve to inspire authors to make suggestions.
I think you should read the works of these authors, and incorporate them into the suggestions at the end of the article.
Reviewer 3 Report
The research topic is interesting, but there are major concerns about the methodology.
First of all, please change the type of the manuscript. This paper is Review, not an "article" (original paper). So please revise this manuscript in line with the guidelines for the review papers.
Please revise the study aim to provide a brief and informative aim that matches the findings presented by the authors.
Please revise the methods section. Inclusion/exclusion criteria; keywords used in the search; timeline (from X to Y) should be precisely described. Some materials can be attached as a supplementary,
Prisma flow chart is unclear. Why only 1 stakeholder was included? How about the government reports and analysis of the national institute of public health or other organizations?
There is a lack of logical structure in the results section. Please revise to provide easy-to-follow text. Please focus on the most important findings rather than put "everything that I have".
Please provide evidence-based conclusions rather than general sentences. Moreover, limitations should be underlined.
Round 2
Reviewer 3 Report
The manuscript was revised in line with the comments from the reviewer. Re-organization of the text allowed to provision of a logical structure of the manuscript that is easy to follow and informative.